# CRAMMING: TRAINING A LANGUAGE MODEL ON A SINGLE GPU IN ONE DAY

## ABSTRACT

Recent trends in language modeling have focused on increasing performance through scaling, and have resulted in an environment where training language models is out of reach for most researchers and practitioners. While most in the community are asking how to push the limits of extreme computation, we ask the opposite question: How far can we get with a single GPU in just one day?

We investigate the downstream performance achievable with a transformer-based language model trained completely from scratch with masked language modeling for a *single* day on a *single consumer* GPU. Aside from re-analyzing nearly all components of the pretraining pipeline for this scenario and providing a modified pipeline with performance close to BERT, we investigate why scaling down is hard, and which modifications actually improve performance in this scenario. We provide evidence that even in this constrained setting, performance closely follows scaling laws observed in large-compute settings. Through the lens of scaling laws, we categorize a range of recent improvements to training and architecture and discuss their merit and practical applicability (or lack thereof) for the limited compute setting.

## 1 SCALING UP AND SCALING DOWN

Large-scale training of machine learning models with transformer architectures has lead to ground-breaking improvements in many sub-fields of natural language processing including language understanding and natural language generation (Vaswani et al., 2017; Dosovitskiy et al., 2021; Radford et al., 2019). The nowadays accepted (but historically surprising) key behavior of these systems is that they reliably *scale* – they continuously improve in performance when the number of model parameters and amount of data grow. These increases in performance are well-described by various power laws as studied by Kaplan et al. (2020). This sets up a dominant paradigm in which scaling is the key to performance improvement (Sutton, 2019).

The power of scale has set off a race to produce extremely large models, which in turn has created an environment where few researchers or practitioners feel that they are capable of training a language model. The original BERT model Devlin et al. (2019), which became a cornerstone transformer for many practical applications in natural language understanding, already required a significant amount of computation to train. Yet, the reproduction and improvements in Liu et al. (2019) further increased its performance by cranking up the level of computation by orders of magnitude. As these pre-trained checkpoints became popular for a range of downstream applications (Wolf et al., 2020), the competition for the largest language model became a focal point for industrial labs. This led to training runs that improved the performance of pretrained language models at the expense of computation at the zettaFLOP scale (Raffel et al., 2020; Yang et al., 2020; Zaheer et al., 2021) and later at the extremely large yottaFLOP scale (Brown et al., 2020; Black et al., 2022; Chowdhery et al., 2022; Rae et al., 2022).

Our goal is to turn this trend on its head and investigate how to best *scale down* language model training and what trade-offs emerge when doing so: *What downstream performance can be achieved by a modest researcher when training from scratch with a single GPU for a single day?* The ability to train a language model to the performance level of BERT with such modest resources has several interesting implications. For one, if scaled-down model pretraining is a viable analogue of large-compute pretraining, then this opens up a host of further academic investigations that

are currently hard to realize for large-scale models. For example, research questions about the differences between existing and new pre-training tasks, tracing model predictions to data points (Ilyas et al., 2022), security questions such as membership inference (Carlini et al., 2022) and data poisoning (Geiping et al., 2021), and a wide range of empirical investigations into topics such as stability or generalization that arise during training (Nagarajan & Kolter, 2019; Jiang et al., 2019). At the same time, we can imagine situations in which legal requirements make it unclear whether models trained on public data with uncertain origin are permissible, and where a practitioner is interested in retraining their language models using a specialized or trustworthy data source (Wilka et al., 2017; Gold & Latonero, 2017).

In addition, we are motivated to benchmark the overall *conceptual* progress of research in this area over the last years, beyond simply turning the scaling knob. The goal of achieving BERT-like performance with modest training resources would have seemed unthinkable in 2018, and yet with modern advances and transformer training techniques this may now be possible.

To answer these questions, we consider a challenge we call "Cramming" – learning a whole language model the day before the test. Our studies begin by investigating many facets of the training pipeline to see which modifications actually improve performance in the scaled-down scenario. We provide evidence that even in this constrained setting, performance closely follows scaling laws observed in large-compute settings. An unsurprising consequence of these laws is that scaling down is hard; while smaller model architectures enable speeding up gradient computations, overall rates of model improvement over time remain nearly constant. Nonetheless, we can find changes to the training pipeline that exploit scaling laws to yield improvements by improving the effective rate of gradient computations without compromising model size. In the end, we are able to train models that achieve respectable performance – often close to and sometimes exceeding BERT on GLUE tasks – on a shoestring budget.

## 2 TYING OUR HANDS BEHIND OUR BACK: A SETUP WITH LIMITED COMPUTE

Before we start this investigation, we want to outline the extent of limitations we are interested in. The rules for cramming are as follows:

- A transformer-based language model of arbitrary size is trained with masked-language modeling, completely from scratch.
- Existing pretrained models cannot be included in any part of the pipeline.
- Any raw text (excluding downstream data) can be included for training. This means that one can achieve speedups by making judicious choices about how and when to sample data, provided the sampling mechanism does not require a pre-trained model.
- The downloading and pre-processing of raw data is exempted from the total compute budget. Pre-processing may include CPU-based tokenizer construction, tokenization, and filtering, but cannot include representation learning (e.g. pre-training a word embedding is not allowed, unless it is counted towards the final runtime).
- Training proceeds on a single GPU for 24 hours.
- Downstream performance is evaluated on GLUE (Wang et al., 2018). Downstream finetuning on GLUE is limited to brief training with only the training data of the downstream task (we consider 5 epochs or less) and needs to work with hyperparameters set globally for all GLUE tasks. Downstream finetuning is excluded from the total compute budget.

In our implementation, we analyze both a setup with a classical `rtx2080ti` GPU (released September 2018) and a separate setup with a more modern `rtxa6000` GPU (released October 2020). We pair each unit with 4 CPU cores and 32GB of RAM.

Why these limitations? We are principally interested in re-investigating the original BERT setup of Devlin et al. (2019) with limited compute. The optimal architecture of the transformer is not fixed, as the optimal size and shape depends on scaling laws (Kaplan et al., 2020). The limitations on usage of existing models rule out distillation from an existing model (Turc et al., 2019; Jiao et al., 2020; Sun et al., 2020; Wang et al., 2020b; Kaliamoorthi et al., 2021) and data filtering based on existing large models (Golchin et al., 2022), both of which ultimately answer questions about compression and

| Group | Target | Accelerator | Time Limit | Total exaFLOP |
|---|---|---|---|---|
| (Devlin et al., 2019) | BERT | 16 `TPU` | 4 days | 680 |
| (Dettmers, 2018) | BERT | 8 `V100` | 11 days | 950 |
| (Narasimhan, 2019) | BERT-large | 1472 `V100` | 47 min | 519 |
| (Raffel et al., 2020) | T5-base | 16 `TPUv3` | 1 day | 170 |
| (Iandola et al., 2020) | squeezeBERT | 8 `Titan RTX` | 4 days | 361 |
| (Narang et al., 2021) | T5 variations | 16 `TPUv3` | 1.75 days | 298 |
| (Tay et al., 2021) | T5-small-L16 | 16 `TPUv3` | 11.2 hours | 82 |
| (Izsak et al., 2021) | BERT variation | 8 `V100` | 1 day | 86 |
| (Liu et al., 2019) | roBERTa-base | 1024 `V100` | 1.25 day | 13 824 |
| (Chowdhery et al., 2022) | PaLM | 6144 `TPUv4` | 50 days | 7 299 072 |
| Our Setup 1 | BERT variation | 1 `rtx2080ti` | 1 day | 5 |
| Our Setup 2 | BERT variation | 1 `rtxa6000` | 1 day | 13 |

**Table 1:** Maximal Throughput available for select training runs of large language models. FLOP Counts for BERT reproductions and related models. Large-scale LMs included only for reference.

transfer of already processed information. Further, we do not want to limit data to the original dataset used to train BERT, wanting to allow for possible improvements through better data curation and quality. The `rtx2080ti` GPU is a natural candidate for this experiment, given that it was released before Devlin et al. (2019), but the more recent `rtxa6000` is also interesting, being arguably the limit of a single-user workstation. At the finetuning stage we want to mimic the original BERT finetuning and evaluation setup, but provide additional limits to prevent gains based on tuning of only the downstream procedure, for example via computationally extensive downstream training (Bahri et al., 2021a), use of multiple downstream datasets (for example continued pretraining with MNLI before finetuning other tasks (Izsak et al., 2021)), and extended hyperparameter optimization for each GLUE task (Devlin et al., 2019; Liu et al., 2019; Lan et al., 2019).

## 3 RELATED WORK ON EFFICIENT TRANSFORMERS

**How long does it take to train BERT?** In general, this question is hard to answer, due to wildly varying hardware and software setups and differing measures of efficiency (Dehghani et al., 2021). An upper bound on the compute of a training run can be established by finding the total number of (low-precision) floating point operations available over the wallclock budget of the run. This peak of total FLOPs in a given time interval is generally not reached in actual compute, even for highly optimized models (Chowdhery et al., 2022), but represents the paid budget required to realize a training run. We summarize budgets for a few select training runs in Table 1. After the original training run for BERT on TPUs, initial reactions estimated up to 11 days of compute for comparable results on GPUs (Dettmers, 2018). However, sustained improvements, especially in software, have reduced the upper limit significantly (You et al., 2019; Narasimhan, 2019). Yet, recipes and implementations generally require entire server nodes (for GPUs) or TPU slices and target larger BERT architectures.

Other work discussing improvements to BERT targets compute settings closer to the original BERT, for example SqueezeBERT (Iandola et al., 2020) employs 8 `Titan RTX` cards for four days. Sellam et al. (2022) note that the original BERT training run is an outlier and doubling its training time more reliably reproduces the original results.

Our central point of comparison for BERT training with limited resources is the work of Izsak et al. (2021) who also attempt the goal of training BERT within 24 hours with overall similar limitations, but use a full server node with 8 `V100` GPUs. Izsak et al. (2021) choose a $BERT_{LARGE}$ architecture variant and train with sequence length of 128, including a range of tweaks such as modified learning rates schedules, large batch sizes, sparse prediction and packed sequences. We re-evaluate this setup as a baseline setting for our own compute budget (which is about 15x smaller).

**Studies of Efficient Transformers** Recent years have seen a flurry of research working to improve and modify the transformer architecture proposed in Vaswani et al. (2017) and we refer to Treviso et al. (2022) for a recent categorization and review of research in this area. Several meta-studies have investigated proposed improvements and modifications: Narang et al. (2021) evaluate a large range of architectural modifications applied to the T5 model pipeline of Raffel et al. (2020) on tasks in both language understanding and translation. The encoder-decoder structure of T5 is closer in spirit

to the original transformer setup, but is understood to behave similarly to BERT when using the encoder component (Liu et al., 2021a). Evaluating modifications with 1.75 days of compute on TPU slices they find that most improvements do not reliably materialize gains in final accuracy. Tay et al. (2021) work in the same setting and evaluate the optimal shape of T5 derived architectures and its relative effects on downstream performance as models are scaled. Further exploration of the scaling behavior of various architectural improvements in Tay et al. (2022a) find that only few modifications outperform the original architecture of Vaswani et al. (2017) at all scales, especially when evaluating downstream accuracy. The meta-study investigating improvements in preparation for extreme-scale training in Scao et al. (2022) focuses on minor modifications to layout, positional embeddings and data sources for autoregressive models, and other extremely-large scale training runs have so far been similarly conservative in their settings (Brown et al., 2020; Black et al., 2022; Rae et al., 2022).

In general though, these evaluations target larger compute settings than we intend to use, and are concerned with whether improvements (often from academic sources and proposed with evaluations on small scales) translate to larger scales. In this work, we set aside the question of (up)scaling and focus only on the limited compute.

**Scaling Laws**   The difficulty in finding tangible improvements is echoed in the scaling laws of Kaplan et al. (2020). Over a wide range of transformer model shapes, Kaplan et al. (2020) find only model size (as number of parameters in non-embedding layers) strongly predicts performance. Further, for a fixed compute budget, an optimal model size can be derived, but performance is only mildly connected to model size - larger models processes less data per unit of compute, but improve faster by almost the same margin. While the precise coefficients and shape of these scaling laws continue to be iterated on (Hoffmann et al., 2022) and adapted for related settings (Bansal et al., 2022; Clark et al., 2022; Bahri et al., 2021b), their overall logic appears hard to escape, even if power laws fit observations somewhat less well on small scales.

## 4 INVESTIGATIONS

For our experimental evaluation we implement and test a considerable number of proposed modifications to the setup of Devlin et al. (2019) for their merits in our limited compute setting as described in Section 2. We first clarify the common implementation and initial data setup, and then investigate architectural, training and dataset improvements.

### 4.1 IMPLEMENTATION DETAILS

We implement everything in PyTorch (Paszke et al., 2017) and to limit our gains from the "software lottery" (Hooker, 2021) we do not use specialized implementations (e.g. as proposed for attention mechanisms in Ivanov et al. (2021); Dao et al. (2022)), which would further bias results towards well-established components. We keep everything on the implementation level of the PyTorch framework, allowing only automated operator fusion (Sarofeen et al., 2022) that can be applied to all components. We run all experiments and ablation studies with the same setup of automated mixed precision (Micikevicius et al., 2018) for standard 16- and 32-bit floating point precision (over full 32-bit float, scaled 16-bit (Rasley et al., 2020) and pure bfloat16 (Wang & Kanwar, 2019). We find no benefit from offloading (Ren et al., 2021; Rasley et al., 2020) in our setting.).

**Initial Data Setup**   We start our investigation with a close analogue to the original raw text sources of Devlin et al. (2019), using a recent dump of the English Wikipedia (20220301.en) and English bookcorpus, noting the commentary of Tan (2019); Bandy & Vincent (2021). We force all text into lower-case, strip accents and non-ascii characters and create an English tokenizer from scratch based only on this data. We choose WordPiece with a vocabulary size of $2^{15} = 32768$ (Wu et al., 2016). We found no significant change in performance with BPE (Sennrich et al., 2016) or SentencePiece with Unigrams (Kudo, 2018; Kudo & Richardson, 2019). Smaller vocabulary sizes ($2^{12}, 2^{13}, 2^{14}$) resulted in worse performance, while larger vocabulary sizes ($2^{16}$) we not reliably better. We pack tokenized data into randomized sequences of length 128 and separate unrelated fragments by <sep> The performance impact from dropping this separator was minimal. No impact was observed from including a <cls> token in pretraining. The shorter sequence length is sufficient for the downstream applications that we are targeting and simplifies attention computa-

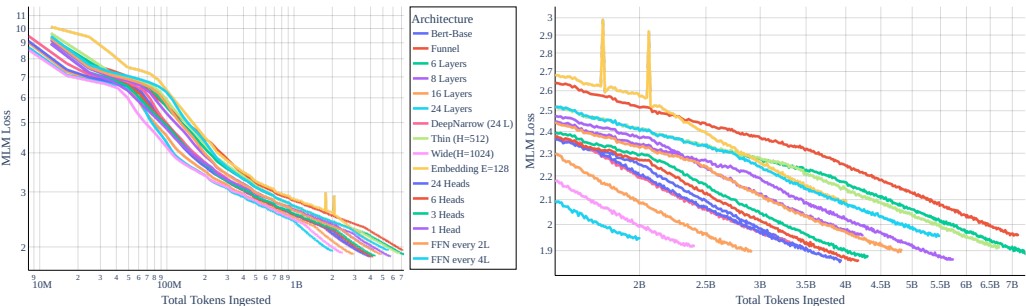

**Figure 1:** Various Transformer architectures and shapes, showing MLM loss versus number of tokens ingested. Left: Global view. Right: Zoom onto 10e8 or more tokens. All models trained with the same budget. We see that improvements through architectural reshaping are minimal; while there are some fluctuations in loss early in training, the rates of loss decay during most of training differ by a multiplicative constant (horizontal shift due to logarithmic horizontal axis) that depends strongly on the model size and not model type.

tions. Packing data into full sequences limits us to simpler sequence losses, but uses the available compute optimally Liu et al. (2019); Izsak et al. (2021). For the targeted compute settings, this sequence length results in micro-batch sizes of 64 to 96 for most variations of the base BERT architecture on `gtx2080ti`, which we will accumulate into larger batch sizes. With our limited compute budget, this produces enough samples to run single-epoch training (Komatsuzaki, 2019; Hernandez et al., 2022) where no data point is revisited.

## 4.2 Modifying the Architecture

The most obvious way to efficiently scale down training is by modifying the model architecture; intuitively, it seems likely that smaller/lower capacity models will be optimal in the cramming regime. In this section, we study the relationship between model type and training efficiency. We see that scaling laws create a strong barrier to scaling down. Per-token efficiency of training depends strongly on model size, but not transformer type. Furthermore, smaller models learn less efficiently, and this largely mitigates any throughput gains. Fortunately, the fact that training efficiency is nearly constant across models of the same size means that we can boost performance by finding architecture modifications that speed up gradient computation while keeping the parameter count nearly constant. This makes architecture selection fairly straightforward as we can make design choices based primarily on how they affect computation time for a single gradient step.

**Scaling laws hold in the low-resource regime** A large corpus of research in recent years has developed architectural improvements to speed up the original transformer. Many of these methods have not been found to improve training for the large-scale T5 architecture Narang et al. (2021); Tay et al. (2022a). But, in the low compute setting where data throughput is of utmost importance, maybe this is the way forward? Scaling laws have been observed by Kaplan et al. (2020) in the high-resource regime, and seem to hold strongly in the limit as resources grow. Surprisingly, these laws also hold in the limit of extreme compute down-scaling, and they create a barrier to low-cost training.

We exemplify the effect of scaling laws for many transformer variants from the literature in Figure 1, where we train each architecture variant with optimized training hyperparameters as described below in Section 4.3. We apply these architecture variants to a shared baseline model that incorporates Pre-Normalization and rotary embedding. Figure 1 visualizes the progress of MLM loss versus the number of tokens ingested in total and all architectures run with the same time budget.

We observe that varying the transformer type and size has only minimal impact on the final loss after 24 hours. Models with more parameters learn more efficiently, as their MLM loss decreases faster on a per-gradient basis. However, smaller architectures make up for their slower learning efficiency by higher throughput, and thus process more tokens over the limited budget. Figure 1 shows that different architectures are unpredictable throughout an initial stage of training (the first 1B tokens), after which the per-token efficiencies differ by only a multiplicative constant (a horizontal shift due to the log axis). This constant depends almost entirely on the model size, not model type, so that all choices reach a MLM loss around 1.9 at the end of training.

**Exploiting the scaling law.** The scaling laws seem to bar us from making large gains via major changes to the transformer size and type, as per-token performance is tightly coupled to model size. As a result, we find no improvements when using a funnel-transformer architecture (Dai et al., 2020; Nawrot et al., 2022), when dropping FFN layers (Sridhar et al., 2022), or when using recurrent layers (Lan et al., 2019), even when trained with BPTT as in Schwarzschild (2021). Rescaling architectures to be deep-narrow (Tay et al., 2021; Wies et al., 2021) provides no gains.

While this principle closes one door for scaling down efficiently, it opens another; Because per-gradient efficiency remains nearly constant for all models of the same size, we can exploit scaling laws by quickly searching for architectural choices that speed up computation while keeping model size roughly constant. A number of obvious optimizations fall into this category, and we describe them below, in addition to several other tweaks that provide marginal but worthwhile/free gains.

**Attention Block:** We disable all QKV biases (Dayma et al., 2021). This exploits the scaling law by removing a layer of computation, making the forward and backward pass somewhat faster, while keeping the model size nearly constant. We find that we can decrease gradient costs by reducing the number of attention heads (Merity, 2019; Araabi & Monz, 2020; Liu et al., 2021b; Javaheripi et al., 2022), as this parallelizes better on the GPU and provides a slight performance boost. We find no benefits from replacements to the softmax operation (Richter & Wattenhofer, 2020). We further keep the original multi-head self-attention mechanism. A large amount of work has been focused on efficient attention (Sukhbaatar et al., 2019; Beltagy et al., 2020; Wang et al., 2020a; Liu et al., 2021c) and studies of efficient attention (Tay et al., 2020a;b). But, because we set the maximal sequence length to 128, attention complexity is less of a concern in our setting. To verify this, we implement the recently proposed FLASH mechanism (Hua et al., 2022), but find no benefits. We further experiment with Fourier attention as proposed in Lee-Thorp et al. (2021), but find no improvements. We supplement the attention with rotary embeddings (Su et al., 2021; Black et al., 2022), which we find to provide small benefits.

**Feedforward Block:** We find empirical gains from disabling all linear layer biases (Dayma et al., 2021). Just as for the attention layers, this leverages the scaling law by accelerating gradient computation without noticeable impacts on model size. As a result, we get higher throughput without compromising the rate at which the model improves. We keep the original feedforward block largely unchanged, finding no benefits from changing to another activation than GELU. We do see small improvements from re-ordering the block into a gated linear unit (Dauphin et al., 2017). In contrast to other work, e.g. (Black et al., 2022), we do not increase the number of parameters in the FFN block to compensate for the halving of the hidden dimensionality due to gating.

**Embedding:** We implement scaled sinusoidal positional embeddings as described in Hua et al. (2022), finding incremental benefits over learned or unscaled sinusoidal embeddings. We see no improvements from decoupling the input and output embeddings (Chung et al., 2020). The suggestion from Lan et al. (2019) to factorize the input embedding provides no gains in our setting. We include a layer normalization at the end of the embedding block.

**Layer Structure:** As observed in many studies, we find that pre-normalization with Layer Norms is beneficial over post Layer Norms (Baevski & Auli, 2018; Xiong et al., 2020). We see no additional benefit from other variants of this modification, such as (Liu et al., 2020b; Shleifer et al., 2021). Further, replacing Layer Normalization with RMS Normalization provides no gains (Zhang & Sennrich, 2019). We note that the key effect of pre-normalization is to stabilize training and enable larger learning rates and reduced warmup, and we see limited benefits from including it by itself. We see no benefits from stochastic dropping of entire layers as described in (Zhang & He, 2020).

**Head Block:** We find that we can remove the nonlinear head without ill effect. We can further drop the decoder bias (Radford et al., 2019) and gain in memory using sparse token prediction (Liu et al., 2019; Izsak et al., 2021). We add a final Layer Norm to stabilize training further.

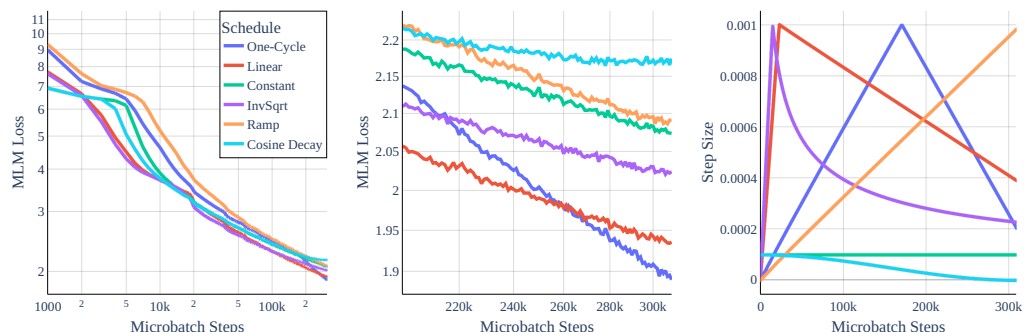

**Figure 2:** Learning Rate Schedules. Although globally many schedule result in similar behavior, we see in the zoom in the middle, that differences do exist. The right side shows the corresponding learning rate schedules.

### 4.3 MODIFYING THE TRAINING SETUP

We study the impact of training hyper-parameters on the BERT-base architecture. The original BERT training recipe understandably results is poor model performance in the cramming setting, and so we revisit a number of standard choices.

**Objective:** We train with only masked language modeling on fully packed blocks of tokens with a masking rate of 15% and the original setup of Devlin et al. (2019) where 10% of all masks are filled with random words and 10% unchanged. We see no improvement from masking at larger rates, e.g. at 40% as proposed in (Wettig et al., 2022). We see no difference enabling or disabling the mentioned 20% rule. We evaluate other functions for the masked-language objective, such as mean-squared error (Hui & Belkin, 2021) or L1 loss, but find no benefits.

**Choice of Optimizer:** We keep Adam (Kingma & Ba, 2015) as the optimizer of choice, with weight decay of 0.01 as described in (Loshchilov & Hutter, 2017), $\beta_1 = 0.9, \beta_2 = 0.98$ and $\varepsilon = 10^{-12}$. We find no noticeable change in varying these parameters in reasonable amounts, e.g. $\varepsilon = 10^{-6}, \beta_1 = 0.9, \beta_2 = 0.999$. We test other first-order adaptive optimizers (Shazeer & Stern, 2018; Liu et al., 2020a) but find no advantages in our setting. We further find no advantages using higher-order optimizers (Yadav, 2020; Anil et al., 2021), but note that especially for higher-order optimizers there is a greater amount of variability in implementation.

**Learning Rate Schedule and Peak:** Following the advice of Izsak et al. (2021), we re-scale the learning rate schedule so that it is tied to our budget and the learning rate decays as the budget reduces to zero. Interestingly, we observe in Figure 2 that while globally a large number of learning rate shapes lead to similar reductions in loss, we find that we can make some gains through the choice of schedule. We find that a simple one-cycle learning rate (Smith & Topin, 2018) with a peak learning rate of $10^{-3}$ leads to minimal pretraining loss within our budget.

**Batch Size Schedule:** A particularity of our setting is that, due to being limited to a single GPU, the micro-batch size that finds its way onto this GPU (96 for most experiments) is several times smaller than the optimal batch size. We find that the optimal batch size in this setting is around 1536 for minimal pretraining loss and 4032 for maximal downstream performance, i.e. we accumulate gradients and only perform an update every 16 and 42 forward/backward passes, respectively.

Fortunately, we can find small speedups by using an aggressive batch size schedule; we increase the number of averaged micro-batches linearly over the course of training. This results in more progress earlier in training, and leads to a small benefit to performance. We also experiment with automatic and adaptive batching rules (De et al., 2017; Bollapragada et al., 2018a;b), but find that the best results from these adaptive schedules resemble the fixed linear schedule. For simplicity we just stick to the simpler linear schedule.

**Dropping Dropout** The original BERT model of Devlin et al. (2019) includes dropout as in Vaswani et al. (2017), which prevents overfitting when training data is small relative to total compute budget. While it can be helpful as a regularizer, dropout effectively reduces the number of gradient

| Dataset | Batch Size | MNLI (m) |
|---|---|---|
| Bookcorpus-Wikipedia | 1536 | 79.8 |
| The Pile | 1536 | 80.5 |
| The Pile (natural data subset) | 1536 | 80.8 |
| C4-Subset | 1536 | 79.1 |
| Bookcorpus-Wikipedia, Deduplication $> 100$ | 1536 | 79.9 |
| Bookcorpus-Wikipedia, Deduplication $> 50$ | 1536 | 79.5 |
| Bookcorpus-Wikipedia, filtered with $t = 0.3$, sorted | 1536 | 80.8 |
| Bookcorpus-Wikipedia, sorted | 1536 | 81.0 |
| C4-Subset, Deduplication $> 100$ | 1536 | 79.2 |
| C4-Subset, filtered with $t = 0.3$ | 1536 | 79.9 |
| C4-Subset, filtered with $t = 0.3$, sorted | 1536 | 81.4 |
| C4-Subset, filtered with $t = 0.3$, larger, sorted | 1536 | 81.9 |
| Bookcorpus-Wikipedia | 4032 | 80.5 |
| C4-Subset, filtered with $t = 0.3$ | 4032 | 82.2 |
| C4-Subset, filtered with $t = 0.3$, sorted | 4032 | 82.5 |
| C4-Subset, filtered with $t = 0.3$ | 8064 | 80.9 |

**Table 2:** Dataset Variations for the optimal model from Section 4.2 and optimal training routine from Section 4.3, modifying final batch size in conjunction with dataset format.

updates seen by each parameter, as updates do not occur when the associated feature is dropped. At the same time, update runtime is not strongly effected by the presence of dropout, and so dropout results in a net reduction in updates per second.

In the cramming setting, training data is large compared to compute. Overfitting is not possible due to the single epoch schedule, and we disable dropout during pretraining (Brown et al., 2020) to maximize the number of parameter updates. We re-enable dropout during downstream fine-tuning with a dropout value of 0.1. Further, we experiment with length curricula (Li et al., 2022) and token dropping (Hou et al., 2022), but find no gains.

## 4.4 Optimizing the Dataset

We found above that scaling laws create a barrier to making major gains (beyond computational efficiencies) with architectural modifications. However, scaling laws do not preclude us from training on better data. Once we have exhausted our ability to train on more tokens per second, we should seek to train on better tokens.

We consider two data based pathways to better down-scaling. First, we can filter, process, or sort the existing data in various ways. Second, we can swap our data source. To this end, we experiment with several subsets of *The Pile* (Gao et al., 2020), containing raw text from only *Gutenberg*, *Books3* and *Wikipedia (en)*. From these Pile datasets we tokenize the first $4 \times 10^6$ entries to generate enough tokens for our single pass. Another popular source of data is C4, the colossal, cleaned version of Common Crawl (Raffel et al., 2020), from which we stream the first $20 \times 10^6$ entries. For each data source we regenerate its own WordPiece tokenizer as described in Section 4.1.

Of these four sources, we find the Pile to perform best in terms of downstream MNLI performance. However, it turns out we can further improve especially the C4 datset through additional processing. We first evaluate deduplication as described in Lee et al. (2022) via exact substring deduplication, but find this not to help in downstream performance in our case. We then test filtering for uncompressible data. We use the tokenizer itself to remove all training sequences from C4 set that cannot be compressed well; we simply set a threshold $t$, e.g. $t = 0.3$, and drop all entries from the dataset where the number of tokens in the entry is larger than $t$ times the number of raw characters. This removes, for example, sequences consisting of hard-to-compress HTML or markdown code. Surprisingly, this results in a measurable improvement on C4, summarized in Table 2.

We then see some further improvements from two directions. First, sorting all tokenized sequences by some metric, and second, increasing the final batch size. For filtering we sort all tokenized sequences by their average (unigram) token prevalence, so that likely sequences occur first. This has some positive effect, and can be strengthened slightly by drawing from a larger cor-

|  | MNLI (m/mm) | SST-2 | STSB | RTE | QNLI | QQP | MRPC | Avg. |
|---|---|---|---|---|---|---|---|---|
| BERT-Base | **83.5**/83.6 | **92.1** | **86.7** | **58.3** | **90.3** | **87.6** | **88.7** | **83.9** |
| BERT-Base(2080ti) | 54.2/54.1 | 80.9 | 14.3 | 52.0 | 59.8 | 65.4 | 78.1 | 57.4 |
| Crammed LM (2080ti) | 82.0/82.5 | 90.4 | 84.9 | 56.5 | 88.2 | 87.0 | 85.0 | 82.1 |
| Crammed LM (A6000) | 83.3/**83.8** | **92.1** | 84.5 | 56.1 | 88.5 | 87.3 | 87.5 | 82.9 |

**Table 3:** Comparison in GLUE-dev performance of baseline BERT to crammed model. Avg. Score is all scores excluding CoLA.

pus, as the unlikely sequences never get reached. Finally, increasing the batch size to 4032 at the end of training (as mentioned in Section 4.3) is disproportionally effective on C4, but less so on `bookcorpus-wikipedia`. We believe that both modifications ultimately reduce the likelihood of training being hindered by fluctuations in the data distribution.

## 5 FINETUNING PERFORMANCE ON GLUE

Finally, we evaluate performance on the GLUE benchmark of Wang et al. (2018), minus WNLI as in Devlin et al. (2019). We note that we only use MNLI (m) during the previous sections and do not tune hyperparameters based on the full GLUE scores. We finetune both the pretrained BERT-base checkpoint and our models under the same constraints laid out in Section 2. For BERT-base, we finetune all datasets for 5 epochs with a batch size of 32 and learning rate of $2 \times 10^{-5}$. For the crammed models, we find that this is not optimal and minor improvements

**Table 4:** Comparison in GLUE-dev performance of baseline BERT to crammed model. Avg. Score is all scores excluding CoLA, GLUE is the full average over the same tasks as in Devlin et al. (2019).

|  | CoLA | Avg. | GLUE |
|---|---|---|---|
| BERT-Base | *56.0* | **83.9** | **80.8** |
| BERT-Base(2080ti) | *2.0* | 57.4 | 51.2 |
| Crammed LM (2080ti) | *41.9* | 82.1 | 77.6 |
| Crammed LM (A6000) | *37.0* | 82.9 | 77.8 |

can be gained from a batch size of 16 and learning rate of $4 \times 10^{-5}$ with cosine decay (this setup does not improve the pretrained BERT checkpoint).

Table 3 and Table 4 describe the performance of this setup on the GLUE downstream tasks (as median over 5 trials). There we compare the original BERT-base checkpoint, a reproduction of the BERT pretraining settings stopped after our budget is reached, and the modified recipe, evaluated with the single day `rtx2080ti` setup and the single day `A6000` setup. Overall, performance is surprisingly decent, especially for the larger datasets of MNLI, QQP, QNLI and SST-2, where downstream finetuning can smooth remaining differences. However, even the smaller datasets mostly work. The average is brought down however by a massive drop on CoLA (corpus of linguistic acceptability) (Warstadt et al., 2019). This behavior is intriguing and we offer two hypotheses. First, it is conceivable that the chosen global hyperparameters for finetuning are a bad fit for CoLA in particular. CoLa performance can be brittle with respect to hyperparameter, with Jiao et al. (2020) training longer only on CoLA or Joshi et al. (2020) training less only on CoLA. Nevertheless, for BERT, a set of global hyperparameters exists, pointing at a deficiency in the crammed model. As a second hypothesis, it is conceivable that these models need to process more text before they memorize enough data to do well on CoLA. This would be in contrast to Liu et al. (2021d) who find that CoLA is learned relatively quickly compared to other downstream tasks when probing intermediate BERT checkpoints. On the other hand, deficiencies on CoLA in particular are also common in approaches that distill BERT into smaller architectures (Sun et al., 2019; Turc et al., 2019; Mukherjee et al., 2021), which might come with limited capacity for linguistic acceptability.

## 6 CONCLUSIONS

We discuss how much performance a transformer-based language model can achieve when crammed into a setting with very limited compute, finding that several strands of modification, such as especially training recipe and data setup lead to decent downstream performance on GLUE. Overall though, cramming language models appears hard, as we empirically find many implications of Kaplan et al. (2020) to still hold in this regime. We hope that this work can provide a baseline for explorations of the question of cramming we formalize in Section 2 and cast a new light on a number of improvements and tricks proposed for transformer architectures in recent years.

REPRODUCIBILITY STATEMENT

We provide code to reproduce all experiments.

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

# A    APPENDIX

# B    LIMITATIONS

In this work, we limited our investigation to transformer-based architectures trained with MLM objectives. However, we do think that the general task of cramming posed in Section 2 is interesting even when relaxing these constraints. There have been a number of modifications proposed to the objective in particular (Joshi et al., 2020; Bao et al., 2020; Bajaj et al., 2022; Tay et al., 2022b). While Artetxe et al. (2022) and Wang et al. (2022) find MLM still to hold up well as a pretraining objective, other suggestions such as ELECTRA (Clark et al., 2019; 2020; He et al., 2021) could be employed which might be beneficial for crammed models. Also, the optimal architecture might not be transformer-based (Merity, 2019; Fusco et al., 2022; Peng, 2021)

# C    OTHER MODIFICATIONS

A few recent developments not included in this study are Roy et al. (2022), Shen et al. (2022), and Mindermann et al. (2022). Modifications further not included in this study are more involved initialization (Zhu et al., 2021), additional objective modifications (Müller et al., 2019), progressive growth (Gu et al., 2021; Shen et al., 2022), convolutional variants (Iandola et al., 2020; Chelombiev et al., 2021; So et al., 2021), sequence recurrence (Lei et al., 2022) and TUPE embeddings (Ke et al., 2020).

# D    ADDITIONAL INFORMATION

| Name | MLM Loss | MNLI-m | MNLI-mm | Tokens/Second |
|---|---|---|---|---|
| Modified Transformer | 1.89 | 81.02 | 81.35 | 50946 |
| DeepNarrow (12 Layers) | 1.94 | 80.90 | 80.97 | 78396 |
| DeepNarrow (24 Layers) | 1.98 | 80.78 | 81.14 | 41289 |
| $E = 128$ | 2.14 | 76.68 | 77.62 | 53267 |
| FFN every 2 blocks | 1.93 | 80.43 | 80.97 | 64774 |
| FFN every 3 blocks | 1.97 | 80.44 | 80.93 | 71634 |
| FFN every 4 blocks | 2.00 | 80.03 | 79.67 | 73319 |
| $H = 512$ | 1.93 | 80.61 | 80.93 | 83718 |
| $H = 1024$ | 1.95 | 80.07 | 80.68 | 32004 |
| 4 Layers | 2.00 | 78.45 | 79.00 | 137127 |
| 6 Layers | 1.93 | 79.49 | 79.82 | 96156 |
| 8 Layers | 1.89 | 81.11 | 81.08 | 74248 |
| 10 Layers | 1.89 | 81.02 | 81.21 | 61431 |
| 16 Layers | 1.92 | 81.39 | 82.10 | 39406 |
| 24 Layers | 2.01 | 80.64 | 80.97 | 26927 |
| Recurrent (1-12) | 2.40 | 77.46 | 77.81 | 52405 |
| Recurrent (2-6) | 2.04 | 80.45 | 80.73 | 53148 |
| Recurrent (3-4) | 2.00 | 80.78 | 81.33 | 51634 |
| Recurrent (4-3) | 1.98 | 80.95 | 81.26 | 51952 |
| BERT-tiny | 3.30 | 56.71 | 57.21 | 914694 |
| BERT-mini | 2.49 | 72.22 | 73.21 | 429593 |
| BERT-Large (Izsak variant) | 2.38 | 76.93 | 77.47 | 13448 |
| Original BERT | 7.54 | 35.45 | 35.22 | 41978 |
| With decoder bias | 1.89 | 80.97 | 81.20 | 51155 |
| $\varepsilon = 6$ in Layer Norm | 1.90 | 80.49 | 81.35 | 51728 |
| Learned Embedding | 1.88 | 80.51 | 81.03 | 52601 |
| No Norm after Embedding | 1.94 | 79.65 | 80.34 | 52175 |
| No Final Norm | 1.89 | 80.40 | 80.89 | 51207 |
| No Skip of Head Transform | 1.88 | 80.49 | 81.19 | 51728 |
| No Rotational Embedding | 1.88 | 80.91 | 81.52 | 53526 |
| Post-LN | 7.54 | 31.82 | 31.82 | 52270 |
| With QKV bias | 1.89 | 80.70 | 80.88 | 51112 |
| With bias in Linear Layers | 1.89 | 80.64 | 81.49 | 50584 |
| 12 Heads | 1.88 | 81.75 | 81.99 | 47967 |

**Table 5:** Additional raw results for experiments considered in the main body. First two blocks: Architectural variants as discussed in Section 4.2. Third block: Ablation study of finally adopted model. All experiments run with the training setup described in Section 4.3 for a day on a single GPU with mixed precision. Batch size is 4032 and dataset is `bookcorpus-wikipedia`. Downstream evaluation as described in Section 5.

