# OpenReview forum: "Cramming: Training a language model on a single GPU in one day"
_ICLR.cc/2023/Conference — Submitted to ICLR 2023_

### Official Review · Reviewer_4joz · 2022-10-25

**Confidence:** 3
**Correctness:** 3
**Technical Novelty And Significance:** 2
**Empirical Novelty And Significance:** 3
**Recommendation:** 6

**Clarity, Quality, Novelty And Reproducibility:**

The paper is mostly well-written with just a few typos. I am not aware of similar investigations in the low-compute regime and believe that many in the community might find this work informative.

**Strength And Weaknesses:**

This work presents a thorough empirical investigation on a topic that is of interest to many researchers who do not have access to large compute clusters. The overall methodology appears to be sound and the final result promising.

However, I am not sure if ICLR is the best venue for this work. There are no new theoretical or algorithmic contributions nor any new insight into representation learning. This does not change my belief that this is an informative paper to many in the community, but it might find a more suitable audience if submitted to venues such as EMNLP. (Disclaimer: I do not regularly publish in this area. I am happy to defer to more experienced reviewers and the AC on this point.)

An additional concern is that it is not clear if the MLM loss is the best metric to track when comparing different model/training design choices. This paper (https://openreview.net/pdf?id=F5uYcwABMu), for example, clearly demonstrates that models with near-identical pretraining loss can perform very differently on downstream tasks due to implicit biases. This should be taken into consideration since many conclusions in this work are based on the MLM loss alone. I encourage the authors to also observe downstream performance and note if they agree or disagree with the MLM loss.

The connection between “cramming” and scaling laws can be clarified. The scaling laws mentioned in this work are empirical observations that a model’s performance strongly correlates its size but not necessarily its shape. The empirical results from this work show that this holds for the low-compute regime, which is somewhat surprising. However, these “laws” are merely empirical observations. It is not clear what the authors mean by “we discuss scaling laws in the low compute regime and find that cramming is hard because the conclusions of Kaplan et al. (2020) so easy to reproduce” (emphasis mine) in the conclusion. It would be better to simply state that the empirical observation from Kaplan et al. holds in the setups investigated, which motivated using architectures that parallelize well. A related concern is that since the dominating factor for performance is the number of FLOPs we can squeeze out of a GPU within a given timeframe, this makes the conclusions of this work somewhat hardware-specific, e.g., they might not hold on TPUs or newer/older GPUs.


**Summary Of The Paper:**

While research on enormous language models have dominated empirical NLP lately, most researchers and practitioners do not have the resources and access to work with these models. This work seeks to answer the empirical question: what is the best performance one can achieve on the GLUE benchmark by training a BERT-like model from scratch on a consumer GPU for one day? The numerous Transformer variants proposed in recent years present another challenge to answering this question – which of these variants are beneficial when one’s compute is extremely constrained?

The authors investigate a wide range of design choices in terms of model architecture, training recipe, and data curation. They also note that the final MLM loss is mostly correlated with the FLOPs spent, not the particular Transformer type and size. This motivates them to choose architectures that parallelize well on GPUs. The final result demonstrates that some combination of the variants proposed in the three years after BERT yields a model that is almost as performant as the original BERT base while using 1/50 of the FLOPs.

**Summary Of The Review:**

Neat empirical investigation with conclusions that might interest many. However, it is not clear if ICLR is the best venue. Moreover, one might argue that MLM loss is not the best criterion to study model and training design choices.

---

> ### Author Response · Authors · 2022-11-19
> **Response to Reviewer 4joz**
>
> Thank you for your helpful review and many remarks. We're glad that you find the question of low-resource training posed in this work to be motivating.
>
> > However, I am not sure if ICLR is the best venue for this work. There are no new theoretical or algorithmic contributions nor any new insight into representation learning. This does not change my belief that this is an informative paper to many in the community, but it might find a more suitable audience if submitted to venues such as EMNLP.
>
> Thank you for this suggestion; we have thought about this, but ultimately think that ICLR is the better venue for this study:
> We do think that our study presents new insights into a very common representation learning task, namely that of language model representation pretraining. At the same time, we do not study language in this paper, really (and only note in passing that we train a model in English on English data sources). We solely focus on studying the machine learning strategies that make this learning task work.
>
> Further, we think this work also doubles as a meta-commentary on many architectural modifications and training routine improvements suggested to improve this task (many discussed at previous ICLR conferences), and the fact that they ultimately have only a limited impact in the scenario we consider.
>
> > An additional concern is that it is not clear if the MLM loss is the best metric to track when comparing different model/training design choices. This paper (https://openreview.net/pdf?id=F5uYcwABMu), for example, clearly demonstrates that models with near-identical pretraining loss can perform very differently on downstream tasks due to implicit biases.
>
> This is an interesting point and discussed well in the concurrent submission you mention. Based on this feedback, we have started to revise our experimental setup to include downstream choices into more evaluations. The dataset choices in Sec.4.4 were already found based on downstream information (and we also find interesting behavior in batch sizes that is only apparent when studying downstream accuracy).
>
> We now include a new table 5 in the appendix, where we note downstream accuracy as well as MLM loss for most of the architectural modifications discussed in Sec. 4.2. Overall though, we have so far not found significant outliers in terms of MLM to downstream performance in this setting, but are running more experiments.
>
> > The connection between “cramming” and scaling laws can be clarified [...] It would be better to simply state that the empirical observation from Kaplan et al. holds in the setups investigated, which motivated using architectures that parallelize well
>
> Thank you for pointing this out. We have revised our conclusions based on your feedback and believe they now reflect our findings in relationship to Kaplan 2020 more accurately.
>
>
> > A related concern is that since the dominating factor for performance is the number of FLOPs we can squeeze out of a GPU within a given timeframe, this makes the conclusions of this work somewhat hardware-specific, e.g., they might not hold on TPUs or newer/older GPUs.
>
> We agree that this a limitation of this work, for example we believe our results to be inapplicable on CPU devices or accelerators like cerebras.
>
> At the same time, we have taken efforts to equalize the implementation of many components, as discussed in Sec. 4.1. As such, we do believe our results would generalize to similar GPU chips, or to the closely related TPU devices. We cannot show this for TPU chips, but for GPUs have done experiments on both the 2018 and 2020 generations, i.e. the 2080ti and rtx a6000.
>
>
>
> Finally, thank you again for your extensive feedback and please do not hesitate to bring up additional questions and comments.

---

> > ### Comment · Reviewer_4joz · 2022-11-22
> > **Response**
> >
> > > Thank you for this suggestion; we have thought about this, but ultimately think that ICLR is the better venue for this study: We do think that our study presents new insights into a very common representation learning task, namely that of language model representation pretraining. At the same time, we do not study language in this paper, really (and only note in passing that we train a model in English on English data sources). We solely focus on studying the machine learning strategies that make this learning task work.
> >
> > Fair enough. I'll leave this to the AC and won't hold it against you.
> >
> > My other concerns are adequately addressed. I have increased the score as a result.

---

### Official Review · Reviewer_xMcb · 2022-10-25

**Confidence:** 3
**Clarity, Quality, Novelty And Reproducibility:** Clarity is clear, however, novelty is…
**Correctness:** 2
**Technical Novelty And Significance:** 1
**Empirical Novelty And Significance:** 2
**Recommendation:** 5

**Strength And Weaknesses:**

Strength:
1. In this paper, several modifications (architecture, training setup and datasets) are explored to check whether there is any improvement. All of these aspects are important and interesting. These can give good insights for the community.

2. Some interesting conclusion are got, for example, training recipe and data setup lead to decent downstream performance on GLUE.

Weaknesses:


1. The investigation of modifications lack convincing experiments. For example, only one task performance (MNLI) is reported for when studying the impact of training hyper-parameters. Other tasks can have a different trend.  And when exploring the effect of the architecture, only MLM loss is report. The performance of downstream tasks can be also important.

2. The technical novelty of this paper is a little limit. The total contributions are also limit.

**Summary Of The Paper:**

In this paper, the paper investigate language model pipeline to see which modifications improve performance in the scaled-down scenario ( a single GPU for 24 hours).

**Summary Of The Review:**

This paper does a lot of Interesting investigations.

---

> ### Author Response · Authors · 2022-11-19
> **Response to Reviewer xMcb**
>
> Thank you for your review and helpful comments. We're happy that you find this submission to be generally insightful for the community.
>
> >  [...] only one task performance (MNLI) is reported for when studying the impact of training hyper-parameters. Other tasks can have a different trend. And when exploring the effect of the architecture, only MLM loss is report. The performance of downstream tasks can be also important.
>
> We do study the final impact on other tasks when considering the full GLUE evaluation in Section 5. Our main goal here was to not over-tune our findings to existing GLUE tasks. As such, we consider only MNLI during our hyperparameter search, and evaluate on all other GLUE tasks (and MNLI-mismatched) without any hyperparameter adjustments. We hope that this will make our findings more robust.
>
> > The total contributions are also limit.
>
> Please let us know which component of our evaluation you would like us to expand upon in our study. We're happy to consider this.
>
>
> Overall, thank you for your time and please let us know if there are more questions or comments.

---

### Official Review · Reviewer_x7E8 · 2022-10-25

**Confidence:** 3
**Correctness:** 3
**Technical Novelty And Significance:** 3
**Empirical Novelty And Significance:** 3
**Recommendation:** 5

**Clarity, Quality, Novelty And Reproducibility:**

Clarity:
- For all the architectural modifications in Sec 4.2, does "no improvement" refer to pretraining loss or downstream tasks?
- In Sec 4.3 batch size schedule, you found optimal performance from different batch size for pretraining (1536) and downstream tasks (4032). Why do you think pretraining loss benefit from smaller batch size? Similarly, could any of the architectural changes in Sec 4.2 have different effects on pretraining vs. downstream?
- For all the changes in Sec 4.2 and 4.3, when you test a specific modification, what were used for the rest of the architecture and training setup? How were they chosen?

Quality:
- The 128 sequence length differs drastically from common choice in language models pretraining (e.g. 1024, or at least 512). To make sure conclusions from this work would apply, some additional experiments with longer sequence length would be helpful.


**Strength And Weaknesses:**

Strengths:
- This paper adds more insights on scaling-down, which is less understood as most concurrent efforts are around scaling-up.
- The experiments have a good coverage in terms of testing various architectural changes from recent literature.
- The final performance on downstream tasks (GLUE) are impressive given the limited compute budget.

Weaknesses:
- The biggest missing piece is an ablation study on the improvement from various architecture changes. In the current version, the authors provided a comprehensive list of things they tried, what helped and what didn't. But it's unknown which change(s) brought the bigger improvement.
- Related to this is the poor performance on CoLA. Although the authors provided several hypotheses, some of them should be tested to verify whether they're actually related to any of the architecture or data changes, or mostly due to reduced model size. For example, one possible cause provided by the authors is that reasonable performance CoLA would need more training data. But are models in these experiment trained on less data compared to BERT-base? Is it possible to see whether the performance gap can actually be closed if the models were trained longer than one day (i.e. seeing more data)?

**Summary Of The Paper:**

This paper investigated pretraining a mask language model in a resource-constrained setting, i.e. a single GPU for one day. The authors empirically tested various architectural and data changes in order to maximize performance. There are some interesting findings, such as per-gradient efficiency only depends on model size, several strategies to filter and sort the training data brought improvement, etc. As a results, the authors were able to push the performance close to BERT base if excluding some outlier tasks.

**Summary Of The Review:**

Overall, this paper tackles an important problem, the experiments design are sound and the empirical findings are informative for language models pretraining. If the authors could add more clarity to generalizability and robustness of the findings, e.g. experiments with sequence lengths longer than 128, further ablate on the causes of drop in CoLA, etc. then the results would be more valuable to language models pretraining.

---

> ### Author Response · Authors · 2022-11-19
> **Response to Reviewer x7E8 (Part I)**
>
> Thank you for your extensive comments and helpful feedback. We're glad that you find this to be an interesting and important problem.
>
> > The biggest missing piece is an ablation study on the improvement from various architecture changes. In the current version, the authors provided a comprehensive list of things they tried, what helped and what didn't. But it's unknown which change(s) brought the bigger improvement.
>
> We now include an ablation study in the appendix, see Table 5 in the revised version. However, we also do want to highlight that none of the architecture variations considered in this study are major improvements - we consider this a key part of our findings and connect to scaling laws as shown in Fig.1. Modifications of the *data* used to train these models in the cramming regime is quantitatively the biggest factor that we found in our investigations.
>
> > Related to this is the poor performance on CoLA. Although the authors provided several hypotheses, some of them should be tested to verify whether they're actually related to any of the architecture or data changes, or mostly due to reduced model size.
> For example, one possible cause provided by the authors is that reasonable performance CoLA would need more training data.  But are models in these experiments trained on less data compared to BERT-base? Is it possible to see whether the performance gap can actually be closed if the models were trained longer than one day (i.e. seeing more data)?
>
> This is actually an interesting question to answer. Ultimately, the models we train on bookcorpus-wikpedia almost see the entire data once. As such, these models have seen the similar training data as BERT-base, but have seen everything only once. Whether training on the same data for a similar number of epochs as BERT-base would recover the same CoLA performance is an interesting suggestion. We are working on investigating this question, but will need more time to scale up to a BERT-sized training run than the discussion period.
>
> > For all the architectural modifications in Sec 4.2, does "no improvement" refer to pretraining loss or downstream tasks?
>
> For all architectural modifications, we considered MLM pretraining loss, where we discarded modifications that noticeably worsened MLM loss, and number of tokens/second for each modification, as a measure of efficiency. However, we have since also added values for MNLI, which we show in the updated Appendix.
>
>
> > In Sec 4.3 batch size schedule, you found optimal performance from different batch size for pretraining (1536) and downstream tasks (4032). Why do you think pretraining loss benefit from smaller batch size? Similarly, could any of the architectural changes in Sec 4.2 have different effects on pretraining vs. downstream?
>
> We think that a smaller batch size may be optimal from an optimization point of view, for similar reasons as pointed out in McCandlish et al., "An Empirical Model of Large-Batch Training". Downstream performance however benefits from the lower variance of a larger batch size as found for roBERTA.

---

> > ### Author Response · Authors · 2022-12-07
> > **Additional Results**
> >
> > As promised, we mentioned that
> > > Whether training on the same data for a similar number of epochs as BERT-base would recover the same CoLA performance is an interesting suggestion.
> >
> > We have now run additional experiments to investigate this question. It turns out that the drop in CoLA performance can actually mostly be explained by the number of heads in the model. In the main body, we adjusted the number of heads to 4, based on pretraining loss and MNLI-m performance, but increasing the number of heads back to 12 improves CoLA performance to 48.79 (compared to 37.9 before).
> >
> > Further, training for 16x the amount of compute (a bit less than half the original BERT budget) then further improves CoLA performance to 50.63, bringing it even closer to original BERT. Comparatively though, the main leverage seems to arise from the number of heads in the model. We will include this information in the next update to our draft.
> >
> > We hope you'll also find this interesting and an answer to your question. Let us know if other questions remain!

---

> ### Author Response · Authors · 2022-11-19
> **Response to Reviewer x7E8 (Part II)**
>
> > For all the changes in Sec 4.2 and 4.3, when you test a specific modification, what were used for the rest of the architecture and training setup? How were they chosen?
>
> For all experiments, we first optimized the training setup as discussed in Sec.4.3 to find optimal hyperparameters for generic crammed model training based on minimal MLM loss. At this stage, we use a generic BERT architecture with minor, well-vetted modifications such as pre-Layer Norm and rotary embeddings. After settling on an optimal training setup, we then test all architecture modifications in Sec 4.2. with this setup. We acknowledge that this is not a complete sweep of all possible variations, but find this to be a reasonable compromise.
>
>
> > The 128 sequence length differs drastically from common choice in language models pretraining (e.g. 1024, or at least 512). To make sure conclusions from this work would apply, some additional experiments with longer sequence length would be helpful.
>
> The conclusions of this work as stated really only apply to models with sequence length 128. We think this is an acceptable trade-off for increased speed, at least for many smaller downstream applications. We do complete the GLUE evaluation with this limited sequence length (truncating longer examples if any exist in a task), and compare to the BERT model trained without this limitation. In this sense, we do think that our evaluation fairly represents the capabilities of the trained model.
> It would be interesting to spend some amount of the 24 hours of training time finetuning the model to a longer sequence length, similar to how the original BERT training trained 90% of the training with a sequence length of 128 and the remainder with a sequence length of 512. At least for the current evaluation setup via GLUE though, we do not believe that this would lead to improvements, even though it is likely to be helpful for other downstream applications.
>
>
> In summary, thank you for your many comments and suggestions. Please let us know if you have more questions or comments!

---

### Official Review · Reviewer_Fi67 · 2022-10-26

**Confidence:** 2
**Correctness:** 3
**Technical Novelty And Significance:** 3
**Empirical Novelty And Significance:** 3
**Recommendation:** 6

**Clarity, Quality, Novelty And Reproducibility:**

The writing is clear and the presentation of issues motivating the current work is adequately articulated in the text. While I am not an expert in the transformer field I feel the authors did a good job explaining the connection between the scaling laws and the downstream performance of the models under consideration. The novelty of the work pertains to the training strategies used to reduce computational costs without removing the total number of model parameters. Although previous works looked at training with limited resources the author's study and extreme training scenario that is likely to be more pertinent and representative of the resources available to typical, non-institutional, researchers.

**Strength And Weaknesses:**

Strengths:
- The motivation for the study proposed in the paper is interesting for a number of reasons. The volume of computation required by many modern transformer models has been prohibitively expensive and therefore out of reach for most researchers for quite a while. By studying the implications of constraining the computational resources on the ability of the model to perform well on certain tasks the authors could provide a way for researchers with limited budgets to participate and utilize these models in fundamentally new ways.
- The trend in the paper to consider modifications that mainly reduce the gradient update cost without significantly impacting the total number of parameters in the model, based on the scaling laws, provides an interesting and unifying theme throughout. The persistence of the scaling laws to influence the performance of the model on tasks is reinforced through empirical evidence throughout and yields interesting insights.
- Performance evaluation on a shoe-string budget of FLOPs compare to other prominent models is impressive.

Weaknesses:
- Similar studies were conducted on a single node with 8 GPUs as noted by the authors. Though that setup had considerably more computational resources the total volume of computation was still a fraction of the amount used by many large research institutions. In light of that work, the scenario presented in this paper may seem somewhat derivative and only marginally interesting.
- It is not clear if or how the observations made in the cramming regime may be used to make more informed decisions regarding the training process in the normal training setting.

**Summary Of The Paper:**

In this paper, the authors study the performance of transformer models on downstream tasks as the total computational budget is decreased.  This process, known as cramming in the paper, turns the problem of training these enormous language models in a new direction from the typical scenario used in industrial labs that train models on a seemingly endless supply of resources. The author's place and exception small limit on the total computation that is allowed to train a transformer model from scratch to the total FLOPs available on a single GPU in 24 hours. By considering the scaling laws of large model transformers the authors mainly investigate training setups that keep the total number of parameters in the model constant but reduce the cost of performing a gradient update. By enumerating a small number of interesting features of the transformer training design space the authors demonstrate that cramming can achieve interesting and sometimes comparable results with larger models using more computation in particular settings and for particular datasets.

**Summary Of The Review:**

Overall I find the motivation for the work and claims made by the authors to be an interesting departure from the traditional language training papers that use exorbitant computational resources. It seems more practical to answer questions about how researchers can do more with less when it comes to allocating resources for training transformer models.

My remarks should be taken with a grain of salt as I am not an expert in this particular area but I would feel more inclined to experiment with transformer models if I felt I could train them to a reasonable level of ability on my modest desktop setup. I believe this sentiment represents the spirit of the paper and the results should be of interest to other members of the research community that are hesitant to participate in this research area because of the perceived computational overheads.

---

> ### Author Response · Authors · 2022-11-19
> **Response to Reviewer Fi67**
>
> Thank you for your helpful comments. We're glad that you, like us, find the question of "cramming" language models interesting.
>
> > Similar studies were conducted on a single node with 8 GPUs as noted by the authors.
>
> We do consider the work of Izsak et al., who describe a single day training run on an 8-GPU server node to be an integral piece of previous work on which we build on. However, the setting considered therein differs in more than only the number of GPUs. Running on a full server node also allows for a much larger model, as significantly more memory is available, especially on the V100 cards considered in Izsak et al. We also experiment with the configuration proposed in Izsak et al., which we can reproduce in our implementation, but find that the reduced compute and memory in our setting lead to the conclusion that this configuration is not well-adapted to the 1 GPU scenario. For example, reaching an MNLI accuracy of 75.21%.
>
> > It is not clear if or how the observations made in the cramming regime may be used to make more informed decisions regarding the training process in the normal training setting.
>
> We think that this is an interesting question. However, we would argue that the cramming regime might be of interest in itself, even if there were no lessons to be learned for normal training settings. That being said, we are evaluating this question, but it will take more time and compute to convincingly study this question than was available during the discussion period, given the additional compute required to scale these results again.
>
> Please let us know if there are other questions!

---

> > ### Author Response · Authors · 2022-12-07
> > **Additional Results**
> >
> > As promised, it would take more time and compute to extrapolate these results to a larger regime, training with this architecture, training setup and data setup for 16x as much compute (208 exaFLOP with an a6000) returns
> >
> > | MNLI-m      | MNLI-mm   | SST2   |STSB |  RTE |  QNLI | QQP | MRPC | CoLA
> > | -----------      | -----------     | -----------      |-----------    | -----------    | -----------    | ------| ------ | ---- |
> > | 86.19         | 86.94          |  93.12        | 86.78       | 56.68      | 91.98 | 88.25 | 89.98 | 50.63 |
> >
> > This model is still trained on a bit less than half of the original BERT compute, but for example beats BERT significantly in MNLI and QNLI tasks (and in in reverse still slightly lagging behing on RTE and CoLA).
> >
> > What is notable here is that this is a direct transfer of the training setup described for the limited budget to the larger budget, with no hyperparameter changes, all described schedulers simply run for the new budget. As such, it appears so far that improvements found in the limited compute setting can be directly translated to larger budgets.
> >
> > We hope this can clarify questions on the applicability of findings for normal training runs. Let us know if you have further remaining questions.

---

### Decision · Program_Chairs · 2023-01-20

**Decision:**

Reject

**Justification For Why Not Higher Score:**

See concerns in metareview.

**Justification For Why Not Lower Score:**

N/A

**Metareview: Summary, Strengths And Weaknesses:**

This paper focuses on pre-training the best-possible (masked) language model on a modest budget (a single day on a single consumer-grade GPU). They consider various modifications to the standard training techniques, including changing the architecture, training setup (e.g. loss function, learning rate schedule, etc.), and data. Congruent with past work, they find that architectural variants make a small difference, whereas changing the training setup and data can be beneficial. Based on these findings, they propose a recipe for producing a BERT model that attains reasonable accuracy on GLUE. While all reviewers appreciated the paper's focus (training a decent MLM on a tiny budget), there were various concerns that could not be fully address in the rebuttal. Specifically, reviewers were interested in a more comprehensive ablation of the different changes introduced in the modified recipe, less of a focus on MLM loss as a target metric, lack of novelty in findings (e.g. it was already known which of the studied components could improve performance), and lack of clarity in terms of what the final recipe was. The consensus is therefore towards rejection.